# HHV-6 Infection and Chemokine RANTES Signaling Pathway Disturbance in Patients with Autoimmune Thyroiditis

**DOI:** 10.3390/v12060689

**Published:** 2020-06-26

**Authors:** Alina Sultanova, Maksims Cistjakovs, Liba Sokolovska, Katerina Todorova, Egils Cunskis, Modra Murovska

**Affiliations:** 1Institute of Microbiology and Virology, Riga Stradins University, LV-1067 Riga, Latvia; maksims.cistjakovs@rsu.lv (M.C.); liba.sokolovska@rsu.lv (L.S.); modra.murovska@rsu.lv (M.M.); 2Institute of Experimental Morphology, Pathology and Anthropology with Museum, Bulgarian Academy of Science, 1113 Sofia, Bulgaria; katerinagencheva@yahoo.com; 3Department of Surgery, Riga East Clinical University Hospital, Clinic “Gailezers”, LV-1038 Riga, Latvia; egils.cunskis@aslimnica.lv

**Keywords:** autoimmune thyroiditis, HHV-6, RANTES, chemokine receptors

## Abstract

The aim of this study was to investigate the role of human herpesvirus-6 (HHV-6) in autoimmune thyroiditis (AIT) development. We examined the possible involvement of HHV-6 gene expression encoding immunomodulating proteins U12 and U51 in AIT development and their role in the modulation of chemokine signaling. One hundred patients with autoimmune thyroiditis following thyroidectomy were enrolled in this study. Nested polymerase chain reaction (nPCR) was used to detect the HHV-6 sequence in DNA samples. Reverse transcription PCR (RT-PCR) with three different HHV-6 gene targets (U79/80, U51 and U12) was to detect active infection markers. HHV-6 load was identified using a commercial real-time PCR kit. Immunohistochemistry was performed to investigate the expression of the HHV-6 antigen and RANTES (Regulated upon Activation, Normal T Cell Expressed and Secreted) in thyroid gland tissue. Different commercial immunosorbent assay kits were used for the detection of RANTES, IFNγ, IL-6, and TNFα levels in the AIT patient group and controls. We detected 98% presence of the HHV-6 genomic sequence in AIT patients’ thyroid gland tissues. Markers of active HHV-6 infection (HHV-6 U79/80, U12 and/or U51 mRNA) were predominant in AIT patients’ thyroid tissue samples in comparison with the control group (56% vs. 6%). Evidence from immunofluorescence microscopy showed that HHV-6 can persist in thyrocytes and can interact with RANTES. Visual confirmation of the intense immunofluorescence signal of RANTES detected in thyroid tissues could indicate high expression of this chemokine in the thyroid gland. On the other hand, immunosorbent assays showed very low RANTES levels in AIT patients’ peripheral plasma. These results indicate that RANTES level in AIT patients could be influenced by HHV-6 activation, which in turn may aid AIT development.

## 1. Introduction

Autoimmune disease affects the thyroid more than any other organ, and the incidence of autoimmune thyroiditis (AIT) has dramatically increased over time worldwide [1]. The most common autoimmune thyroid diseases are Hashimoto thyroiditis (HT) and Graves’ disease. Like other autoimmune diseases, AIT predominantly affects women, which could be explained by differences in hormonal background [2]. Autoimmune thyroiditis (AIT) is a big challenge for researchers, as many of the etiological factors are still unknown. 

Recently, more and more attention has been given to the possible involvement of viruses in the development of different autoimmune diseases. AIT is no exception—some studies bring evidence of human herpesvirus-6 (HHV-6) involvement in this disease’s development. A study mapping the reservoirs of human herpesviruses detected EBV, CMV and HHV-6 in the thyroid gland tissues of eight autopsies, although without detecting active infection, showing that the thyroid is potentially susceptible to herpesvirus infection. [3]. A study published by Italian researchers links HHV-6 to HT [4]. Authors demonstrated that HHV-6 infection could be detected more frequently in thyroid fine needle aspirates (FNA) from HT patients than in a control group (82% and 10%, respectively). Furthermore, active HHV-6 transcription is observed only in HT thyrocytes, compared with the latent infection found in HHV-6-infected thyroid glands used as a control. These researchers propose a potential mechanism for HHV-6-induced autoimmunity, demonstrating that follicle cells infected with HHV-6 become susceptible to NK-mediated killing [4]. Our previously published data also showed an almost 100% incidence rate of the HHV-6 genomic sequence in thyroid gland tissue samples acquired from AIT patients’ post-surgical materials [5]. We found a significantly higher presence of the HHV-6 activation marker (HHV-6 U79/80 mRNA) in AIT patients in comparison to controls (18/44 (41%) vs. 1/17 (6%), respectively; *p* = 0.0118) [5].

After primary infection, HHV-6 establishes life-long persistency. HHV-6 has the ability to infect almost every cell in the organism using the CD-46 receptor [6]. Moreover, HHV-6 is known to have strong immunomodulating properties, which could affect the host over a long period of time and contribute to several autoimmune disorders, including autoimmune hemolytic anemia/neutropenia [7], autoimmune acute hepatitis [8], multiple sclerosis [9,10,11].

Chemokines and their receptors are crucial in both innate and adaptive immune responses. They ensure the migration of immune cells, their activation and differentiation, which is why many viruses have evolved the ability to encode viral chemokine receptor homologs. Human chemokine receptors are G-protein coupled receptors (GPCR). They possess certain structural and chemical characteristics, like the seven transmembrane domains, which have also been described for viral GPCRs [12,13].

HHV-6 has two genes (U12 and U51) that encode putative homologs of cellular G-protein-coupled receptors (GPCR) [14,15]. Both share some sequence identity with human CCR 1, 3, 5, 7 and 10 and can bind several chemokines, among them the chemokine RANTES (Regulated upon Activation, Normal T Cell Expressed and Secreted) [12]. It has been shown that proteins which are encoded by HHV-6 U12 and U51 genes could be expressed on the surface of epithelial and some peripheral blood mononuclear cell populations, which makes them a potential cause for evoking autoimmunity by making the hosts’ GPCRs into targets for autoreactive T and B lymphocytes [16,17]. Additionally, it has been shown in vitro that HHV-6 GPCR homologs could interact with cytokines’ signaling pathway by down-regulating RANTES expression and secretion. RANTES is a β chemokine, which can induce leukocyte migration upon binding to CCR1, 3 or 5. RANTES is mainly produced by CD8+ T cells, epithelial cells, fibroblasts and platelets. Virus-specific CD8+ T cells release RANTES, demonstrating its role in antiviral immune responses. Abnormal levels of RANTES have also been linked to multiple inflammatory diseases and several autoimmune diseases, but a clear connection between AIT and RANTES levels has not been established [18,19]. 

Although multiple studies suggest the association of HHV-6 and AIT, the mechanisms by which the virus triggers or aggravates autoimmunity have not been clearly elucidated. HHV-6-encoded chemokine receptor homologs are very poorly studied, so their role in disease development is unclear. The aim of this study was to investigate the role of HHV-6 by examining the possible involvement of HHV-6-encoded immunomodulating proteins U12 and U51 in the development of AIT and their role in chemokine signaling modulation.

## 2. Materials and Methods 

### 2.1. Study Groups

One hundred patients with autoimmune thyroiditis following thyroidectomy were enrolled in this study, of which 6 were males (6%) and 94 were females (94%), with a median age of 52 (interquartile range [IQR]: 42–61). 

The control group included 30 autopsied subjects (26 women and 4 men; median age 58; IQR: 51–67) without thyroid pathologies (data on the frequency of HHV-6 infection markers were taken from previous studies [5] and additional research was carried out to investigate HHV6 U12 and U51 gene expression) and 35 healthy blood donors (30 women and 5 men; median age 39; IQR: 31–48) to compare serology results, as the plasma samples from autopsied subjects were not available.

Permission to conduct the research was received from the Ethics Committee of Riga Stradins University (ethical code Nr 67 and date of approval 25 October 2012) and written consent was obtained from all the patients and relatives, respectively. Tissue samples were received from the Riga East Clinical University Hospital.

### 2.2. Nucleic Acid Isolation, Complementary DNA (cDNA) Synthesis and Quality Determination

DNA from thyroid gland tissues and peripheral blood was extracted using the phenol–chloroform method. RNA from thyroid gland tissue and peripheral blood mononuclear cells (PBMC) specimens was extracted using TRI reagent (Life Technologies, Carlsbad, CA, USA), and cDNA was synthesized using an innuSCRIPT One Step RT-PCR SyGreen Kit (Analytik Jena, Jena, Germany). The integrity and quality of isolated RNA was tested in denaturing gel electrophoresis using NorthernMax™-Gly Gel Prep/Running Buffer according to the manufacturer’s protocol (Thermo Fisher Scientific, Waltham, MA, USA). The quality of genomic DNA and synthesized cDNA was determined by beta (β)-globin polymerase chain reaction (PCR) with appropriate primers [20]. 

### 2.3. Detection of HHV-6 Genomic Sequences and Gene Expression Using Nested PCR (nPCR)

The nPCR technique was used to detect viral genomic sequences in DNA isolated from thyroid gland tissue and whole blood. PCR amplification of viral DNA was carried out in the presence of 1 µg of DNA. HHV-6 was detected in accordance with Secchiero et al., 1995 [21], with primers targeting the HHV-6 *major capsid protein* (MCP) gene. Positive controls (HHV-6A and HHV-6B genomic DNA; Advanced Biotechnologies Inc, Columbia, MD, USA) and negative controls (DNA obtained from healthy HHV-6-negative blood donors and without template DNA) were included in each experiment.

To identify active HHV-6 replication, HHV-6 U79/80 gene expression was detected using nPCR with cDNA as the template [16] because U79/80-gene-coded proteins have a role in viral DNA replication [22].

To determine HHV-6 U12 and U51 gene expression, nPCR was performed with cDNA as the template [16].

### 2.4. HHV-6 Load Determination Using Quantitative PCR

AIT patients’ and autopsied subjects’ thyroid tissue DNA samples, which were positive for the presence of the HHV-6 genome sequence, were used for HHV-6 load detection using the HHV-6 Real-TM Quant (Sacace Biotechnologies, Como, Italy) commercial kit in accordance with the manufacturer’s instructions (target region: HHV6 polymerase gene; sensitivity: <5 copies of HHV6 DNA per 10^5^ cells. 

### 2.5. Determination of RANTES (CCL5) Level in Peripheral Blood Plasma by ELISA

Plasma samples acquired from AIT patients’ and blood donors’ peripheral blood were tested for RANTES level using an ELISA commercial kit (ab100633, Abcam, Cambridge, UK). The ELISA kit was used in accordance with the manufacturers’ instructions (sensitivity < 3 pg/mL; detection range 2.74 pg/mL–2000 pg/mL).

### 2.6. Determination of IFNγ, IL-6, TNFα and RANTES (CCL5) Levels in Peripheral Blood Plasma by Suspension Multiplex Immuno Assay (SMIA)

Plasma samples acquired from AIT patients’ and blood donors’ peripheral blood were tested for IFNγ, IL-6, TNFα and RANTES levels using an SMIA commercial kit (HCYTOMAG-60K, Millipore, Burlington, MA, USA). The SMIA kit was used in accordance with the manufacturers’ instructions (sensitivity for RANTES: <1.2 pg/mL, for other cytokines <0.9 pg/mL).

### 2.7. Immunofluorescence Labelling of Formalin-Fixed Paraffin-Embedded (FFPE) Thyroid Tissues

For this study, FFPE thyroid tissue slides (4–5 µm sections) were acquired, the majority of which were from Grave’s disease and Hashimoto thyroiditis patients.

Slides were loaded into glass slide holders and dewaxed as follows: twice in 100% xylene (Fisher Scientific, Loughborough, Leicestershire, UK) for 15 min. Afterwards, rehydration was done by putting slides once into 96% ethanol for 10 min, following a 70% ethanol change for another 10 min. Finally, the slides were washed three times with PBS (pH = 7.4) for 5 min. A PAP pen for immunostaining (Z672548-1EA, MilliporeSigma, St. Louis, MO, USA) was used to ring the tissue section. 

Cell permeabilization with PBS Triton X100 0.1% for 15 min was necessary, as it improved immunostaining quality. After washing three times with PBS (pH = 7.4) for 5 min, 200 µL of blocking solution (PBS plus 1% bovine serum albumin (A3059, Sigma, Germany) and 2% foetal calf serum (Invitrogen, Carlsbad, CA, USA), filtered through a 0.2 µm filter) was added to the samples and placed in a moist chamber at room temperature for 1 h. After blocking, slides were washed three times with PBS.

For immunostaining, mouse monoclonal antibody against HHV-6 gp82/105 antigen (dilution 1:100, sc-65448, Santa Cruz, Dallas, TX, USA) and goat polyclonal anti-RANTES (CCL5) antibodies (dilution 1:100, ab10590, Abcam, Cambridge, UK) were used as primary antibodies. Rabbit Anti-Mouse IgG Alexa H&L Fluor^®^ 488 (dilution 1:200, ab150125, Abcam, UK) and Donkey Anti-Goat IgG H&L Alexa Fluor^®^ 647 (dilution 1:200, ab150131, Abcam, Cambridge, UK) were used as secondary antibodies. All antibodies were diluted in staining buffer (PBS 1% BSA). Incubation with primary antibodies was done overnight in a moist chamber at +4 °C; after three washes with PBS for 5 min, slides were incubated with secondary antibodies in a moist chamber at room temperature for 1 h. After another three washes with PBS, slides were incubated with PBS containing 1.43 µM 4′,6-diamidino-2-phenylindole (DAPI; D21490, Invitrogen, Carlsbad, CA, USA) for five minutes at room temperature. BrightMount/Plus Aqueous Mounting Medium (Anti-fading) for Fluorescent Staining (ab103748, Abcam, Cambridge, UK) was added to the slides and sealed with a cover glass and nail varnish all around the perimeter. For immunofluorescence sections, digital images were captured using a Nikon EclipseTi-E confocal scanning microscope.

## 3. Results

### 3.1. Detection of HHV-6 Virus Genomic Sequence and Its mRNAs in AIT Patients and Control Group Autopsy Specimens

The HHV-6 genomic sequence was found in 98 (98%) AIT patients’ thyroid gland tissue DNA samples and in 23 (77%) autopsy samples (*p* = 0.0005) (data on the control group taken from previous publication by Sultanova et al., 2017 [5]). The HHV-6 genomic sequence was found in only 16 out of 98 (16%) AIT patients’ peripheral blood DNA samples, showing that the infection occurs predominantly in the thyroid gland, not in the circulating cells.

The overall presence of different HHV-6 mRNAs (U79/80, U12 and/or U51) was found to be significantly higher (*p* = 0.005) in the AIT patient group than in the control group (56% against 6%, respectively). The presence of HHV-6 chemokine receptor mRNAs (U12 and/or U51 mRNAs) was found only in AIT patients’ thyroid gland tissue samples (38/98; 39%).

As shown in Figure 1:17/98 (17%) AIT patients’ thyroid tissue samples and 8/16 (50%) PBMC samples were positive for HHV-6 U79/80 mRNA.6/98 (6%) AIT patients’ thyroid tissue samples were positive for HHV-6 U12 mRNA, while none of the PBMC samples were HHV-6 U12 mRNA-positive and 3/98 (3%) thyroid tissue samples of AIT patients and 4/16 (25%) PBMC samples were positive for HHV-6 U51 mRNA.U79/80 + U12 mRNA was found in 3/98 (3%) AIT patients’ thyroid tissue samples, while none of the PBMC samples were HHV-6 U79/80 + U12 mRNA-positive; U79/80 + U51 mRNAs were present in 3/98 (3%) AIT patients’ thyroid tissue samples and in none of the PBMC samples.U79/80 + U12 + U51 mRNAs were present in 19/98 (20%) AIT patients’ thyroid tissue samples and in none of the PBMC samples.U12 + U51 mRNAs were found only in 4/98 (4%) AIT patients’ thyroid tissue samples.The presence of HHV-6 U79/80 mRNA was only detected in 1/17 autopsy samples (data from previous study [5]), and no HHV-6 U12 or U51 mRNAs were found in this study.

Presence of any HHV-6 transcripts (U79/80; U12 and U51 mRNAs) in individuals was considered as the presence of an active viral infection, while absence was considered as the presence of a latent infection, as these genes are expressed during productive viral infection.

### 3.2. HHV-6 Load in Thyroid Tissue Samples

Median HHV-6 load was found to be higher in AIT patients’ thyroid gland tissue samples with markers of active HHV-6 infection (582.7 [IQR: 168.4–2191.0] viral copies/10^6^ cells) in comparison to AIT patients’ thyroid gland tissue samples without markers of active HHV-6 infection (361.0 [IQR: 110.0–1029.0] viral copies/10^6^ cells); however, a Mann–Whitney test showed no significance (*p* = 0.3851; Figure 2).

Comparison of HHV-6 loads found in patients’ thyroid gland samples with HHV-6 U12 and/or U51 mRNA expression and viral load in thyroid tissue samples without HHV-6 U12 and/or U51 mRNA expression showed a significantly higher median value in the former (1011.0 [IQR: 377.7–3076.0] vs. 425.6 [IQR: 64.8–2385.0] viral copies/10^6^ cells; *p* = 0.0494; Figure 2).

Samples where at least one of U12, U51 or U79/80 mRNA was detected were considered to have active HHV-6 infection, whereas, when no viral mRNAs were detected, samples were considered to harbor latent HHV-6 infection. Information about HHV-6 load and gene expression was acquired using qPCR and nPCR methodology, respectively.

### 3.3. RANTES Level in Plasma of AIT Patient and Blood Donor Groups Detected with ELISA and SMIA

RANTES plasma levels were determined by ELISA in patients with AIT and blood donors. AIT patients had statistically lower RANTES levels (median 9.267 [IQR: 3.489–16.93] pg/mL) than blood donors (median 56.6 [IQR: 30.71–101.5] pg/mL). The same samples were tested additionally with a commercial SMIA kit. RANTES plasma levels determined by SMIA in patients with AIT and blood donors also showed that AIT patients had statistically lower RANTES levels (median 150.3 [IQR: 71.6–418.2] pg/mL) than blood donors (median 1359.0 [IQR: 844.2–2596.0] pg/mL; Figure 3) but with better detection range and higher sensitivity.

AIT patients with active infection, in which U12 and U51 gene expression was detected, showed much lower RANTES levels than patients with latent viral infection (median 45.58 [IQR: 3.2–158.2] vs. 108.2 [IQR: 28.0–306.9] pg/mL); however, significance was not established (*p* = 0.0607).

### 3.4. IFNγ, IL-6 and TNFα Levels in Plasma of AIT Patient and Blood Donor Groups

IFNγ and TNFα levels were found to be significantly (*p* = 0.035 and *p* = 0.007) elevated in AIT patients’ plasma samples (median 2.7 [IQR: 2.1–3.0] and 14.0 [IQR: 10.1–16.9] pg/mL, respectively) in comparison to control group samples (median 2.2 [IQR: 1.7–2.7] and 10.6 [IQR: 9.5–13.5] pg/mL; Figure 3). Although some AIT patients had very high IL-6 levels, the median value was not found to be significantly different from the control group’s median (median 9.1 [IQR: 1.1–76.2] vs. 7.3 [IQR: 2.6–21.5] pg/mL; Figure 3). The absence of statistical significance could be explained by the extremely low IL-6 level in individuals, as detectable levels were found only in 10 AIT patients’ and in 8 control groups’ plasma samples (assay sensitivity is 0.9 pg/mL; HCYTOMAG-60K, Millipore, USA).

### 3.5. Detection of HHV-6 Antigen and RANTES by Immunofluorescence Microscopy 

Immunostaining with antibodies against HHV-6 gp82/105 antigen and goat polyclonal anti-RANTES (CCL5) antibodies showed a clear presence of viral infection in thyrocytes (thyroid epithelial cells forming follicles) (Figure 4; Row A—Alexa488 and Merged (arrows)). The histological structure of the gland was expressed in a single layer of well-distinguished round nuclei of thyroid follicular cells lining the colloid matrix. In some areas, rare, smaller lymphocytic nuclei were seen, integrated between the cells of the epithelial cell layer forming follicles and separated by interstitium. Morphological investigations of all samples revealed the localized presence of HHV-6 antigen, often accompanied by RANTES expression). Against the background of the blue fluorescence-marked (DAPI-stained) nuclei of the epithelial cells encircling the colloid pools of the follicular cluster and the cells within the connective intra-lobular tissue, a weak fluorescence signal of HHV-6 antibody tagging was observed in separate cells. A mainly zonal distribution of HHV-6 viral particles in the apical cell cytoplasm was observed (Figure 4; Row A—Alexa488 and Merged (arrows)). In some areas, a red fluorescence signal was found indicating the presence of RANTES, mainly observed in close connection to the epitopes of viral expression marked with Alexa488 (Figure 4; Row B—Alexa647 and Merged (arrows)). Another important point is that the simultaneous presence of HHV-6 antigen and RANTES was observed at the site of the damaged cells’ structure (Figure 4; Row B—Merged and TD + Merged). Meanwhile, in undamaged cells, only the viral antigen was detected (Figure 4; Row A—Merged and TD + Merged). This could indicate the important role of RANTES in the destruction of thyroid cells and possible involvement of this chemokine in interaction with HHV-6.

No presence of the HHV-6 antigen or RANTES was observed in the control patient (thyroid gland adenoma) sample (Figure 4; Row C—Alexa488 and Alexa647). Additionally, the staining control showed the absence of secondary antibodies’ unspecific binding (Figure 4; Row C—Alexa488 and Alexa647).

## 4. Discussion

The etiopathogenesis of AIT is not yet fully understood as the disease is multifactorial. Nowadays, there is more concern about the involvement of viral infections as important factors in AIT development. In this study, we have shown evidence of the importance of HHV-6 in AIT development. We detected almost 100% presence of the HHV-6 genomic sequence in AIT patients’ thyroid gland tissues. This is a significant difference in comparison with HHV-6 distribution in the control group (98% vs. 77%; *p* = 0.0058) published in our previous study [5]. Markers of active HHV-6 infection (HHV-6 U79/80, U12 and/or U51 mRNA) were predominantly present in AIT patients’ thyroid tissue samples in comparison to the control group (56% vs. 6%). These results indicate that HHV-6 cannot be ignored as a potential autoimmunity trigger factor in AIT development, as its presence and gene expression was predominantly found in AIT patients.

Infectious agents like HHV-6 can initiate autoimmune processes in the thyroid gland by a variety of mechanisms, such as modifying the antigen itself, mimicking the host’s molecules, and initiating polyclonal T-cell activation [23]. HHV-6 encodes two viral chemokine receptors (GPCRs), U12 and U51, which are structurally similar to cellular G-protein-coupled receptors [15], but the role of these genes is not well understood. HHV-6 U12 and U51 are functionally similar to the much-better-studied CMV protein US28 (pUS28), which enhances the course of CMV infection by inhibition of chemokines through interactions the with host’s beta-chemokine receptors [24]. Proteins encoded by the HHV-6 U12 and U51 genes can be expressed on the surface of epithelial and some peripheral blood mononuclear cell populations, making them a potential cause for the initiation of the autoimmunity process, where host GPCRs could be targeted by auto-reactive T and B lymphocytes. In-vitro HHV-6 GPCR may also interact with the cytokine signaling pathway, leading to down-regulation of RANTES [14,16]. Both of these factors make the U12 and U51 proteins ideally suited for studying the involvement of HHV-6 in the development of autoimmune thyroiditis.

In this study, 39% of AIT patients had HHV-6 U12 and/or U51 mRNA in thyroid tissue samples. This means that more than one third of AIT patients’ thyroid tissue expressed viral chemokine receptors and none of the control group’s thyroid samples contained any of these mRNAs. Additional experiments are needed to obtain evidence for the presence of HHV-6 proteins on thyrocytes by methods such as Western blotting or immunofluorescence. However, specific antibodies that are not currently available on the market are needed to obtain clearer evidence. The lack of these antibodies could be explained by the difficulty in obtaining antigens for immunization because of the transmembrane nature of these proteins and by the difficulty in purifying them. This problem could be solved by the use of many different synthetic peptides (made from HHV-6 amino acid sequences).

For the next step, we determined RANTES levels, as HHV-6 U12 and U51 can interfere with this chemokine, and the chemokine plays an important role in the selective recruitment of circulating T-cells. Initially, we used a commercial ELISA kit (ab100633, Abcam, UK) and received results that showed statistically lower RANTES levels in the AIT patients’ group (median 9.267, IQR: 3.489–16.93 pg/mL) compared to blood donors (median 56.6, IQR: 30.71–101.5 pg/mL). However, we soon switched to a different immunological assay—the Suspension Multiplex Immuno Assay (SMIA) and Luminex 200 system. SMIA has two major advantages in comparison to ELISA: (1) since the reaction runs in suspension, it has a higher sensitivity; (2) MagPlex—magnetic beads with 100 different specific fluorescent regions—allow simultaneous detection of up to 100 different analytes in a sample. The SMIA kit showed similar results to ELISA but with a higher range for a significantly lower level of RANTES in AIT patients’ plasma samples (median 150.3 [IQR: 71.6–418.2] pg/mL) as opposed to blood donors (median 1359.0 [IQR: 844.2–2596.0] pg/mL). This range is more realistic in comparison with other published results on RANTES [25,26]. At the moment, it is not clear how low RANTES production is involved in AIT development; however, in studies of other autoimmune diseases—autoimmune Addison disease and rheumatoid arthritis—RANTES plasma level was significantly elevated [27,28]. One of the studies showed that AIT patients could be more susceptible to lower RANTES levels as they had a significantly lower frequency of the RANTES 403A genotype, which is involved in higher production of RANTES in comparison to controls [29].

Findings from immunofluorescence microscopy show that HHV-6 can persist in thyrocytes and can interact with RANTES. Visual confirmation of an intense immunofluorescence signal of RANTES detected in thyroid tissues could indicate higher expression of this chemokine in the thyroid gland—however, quantitative RANTES gene expression comparison in peripheral blood and thyroid tissue should be performed in order to obtain stronger evidence.

In addition, were reported data suggesting a potential role of follicular cells, through the production of the chemokines Ip-10, Mig and RANTES, in activating specific lymphocyte subsets in AIT patients [30]. Therefore, significant differences in the expression of RANTES between thyroid gland follicular cells and peripheral blood can create opportunistic circumstances for lymphocytes’ migration into the site of higher expression of this chemokine, i.e., the thyroid gland, thereby provoking autoimmunity processes. This means that any infection in the thyroid gland, in this case HHV-6 infection, could possibly act as a trigger factor for the entire cascade leading to AIT development.

Moreover, in this study, we found that levels of INF-γ and TNF-α, pro-inflammatory cytokines which are implicated in RANTES induction, were significantly elevated in AIT patients in comparison to controls. These pro-inflammatory cytokines stimulate endothelial cells to express the CC chemokine RANTES, which leads to selective recruitment of circulating cells [31]. Elevated INF-γ and TNF-α levels and downregulated CC chemokine (RANTES levels are similar to classic β-herpesvirus infection and its mechanism of evading the immune response. Induction of INF-γ and TNF-α occurs before viral gene expression, suggesting that the response might be a cellular response to announce foreign invasion and to stimulate an inflammatory response, but, after expression of viral genes begins, down-regulation of RANTES and other CC chemokines takes place [31,32,33,34]. Although statistical significance was not found, we observed lower RANTES levels in AIT patients in whom U12 and U51 gene expression was detected (median 45.58 [IQR: 3.2–158.2] vs. 108.2 [IQR: 28.0–306.9] pg.

Furthermore, the highest median viral load was detected in AIT patients with lowest RANTES level and with HHV-6 U12/U51 mRNA. These results indicate that RANTES level in AIT patients could be influenced by HHV-6 activation, which in turn may aid AIT development. In conclusion, to make strong statements, a deeper investigation should be done, although, from our results, it is clear that AIT patients have a disturbance in the RANTES signaling pathway, and that HHV-6 could influence it.

## Figures and Tables

**Figure 1 viruses-12-00689-f001:**
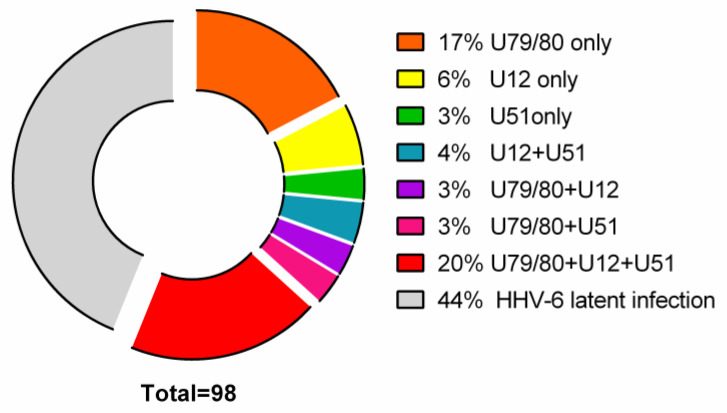
Distribution of HHV-6 U79/80 (encoding proteins involved in viral replication), U12 and U51 mRNAs (late and early genes, respectively, encoding viral homologs of cellular G-protein-coupled receptors) in AIT patients’ thyroid gland tissue samples. Results from nPCR investigations of viral gene expression using cDNA as a template.

**Figure 2 viruses-12-00689-f002:**
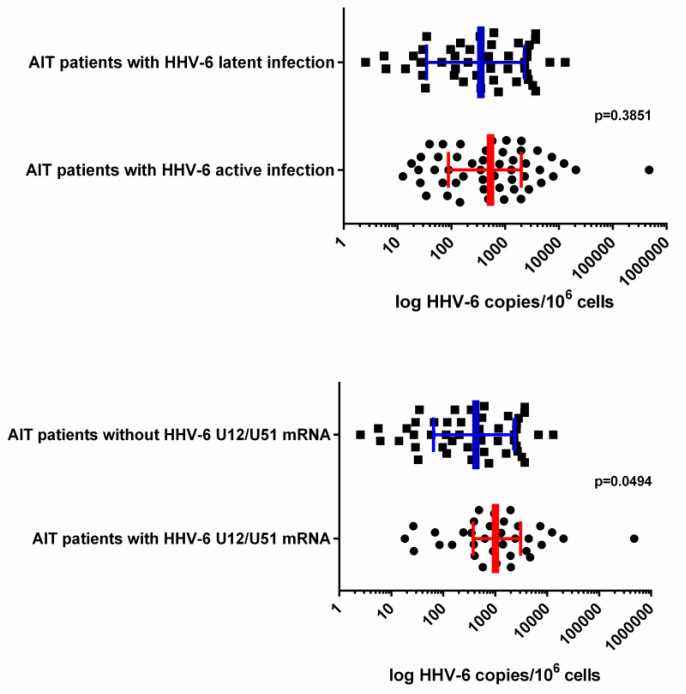
Median HHV-6 load (error bars indicating IQR) detected in AIT patients’ thyroid gland Table 6. Load was indicated as log HHV-6 copies per 10^6^ cells. HHV-6 U12 and U51 encode viral homologs of cellular G-protein-coupled receptors.

**Figure 3 viruses-12-00689-f003:**
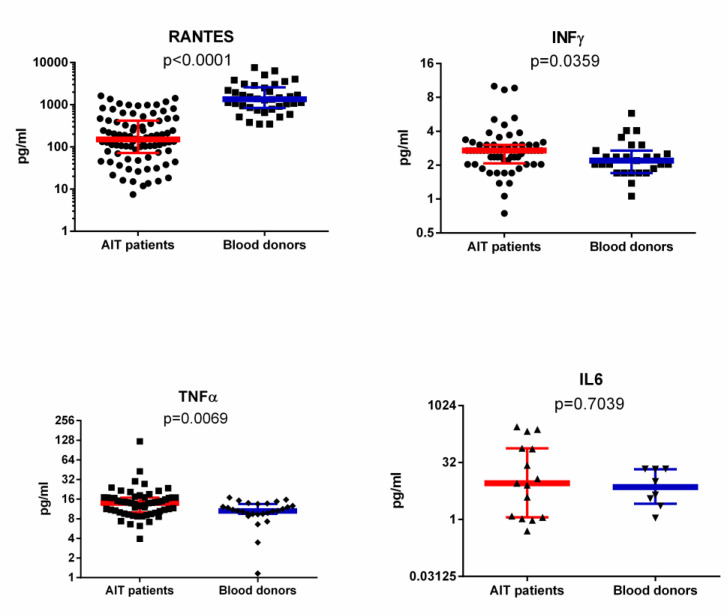
RANTES, IFNγ, IL-6, and TNFα median levels (error bars indicating IQR) in the plasma of AIT patient and blood donor groups. Chemokine (RANTES) and pro-inflammatory cytokine (IFNγ, IL-6 and TNFα) levels from SMIA investigations are expressed as pg/mL. IFNγ, IL-6 and TNFα were chosen as they participate in RANTES induction.

**Figure 4 viruses-12-00689-f004:**
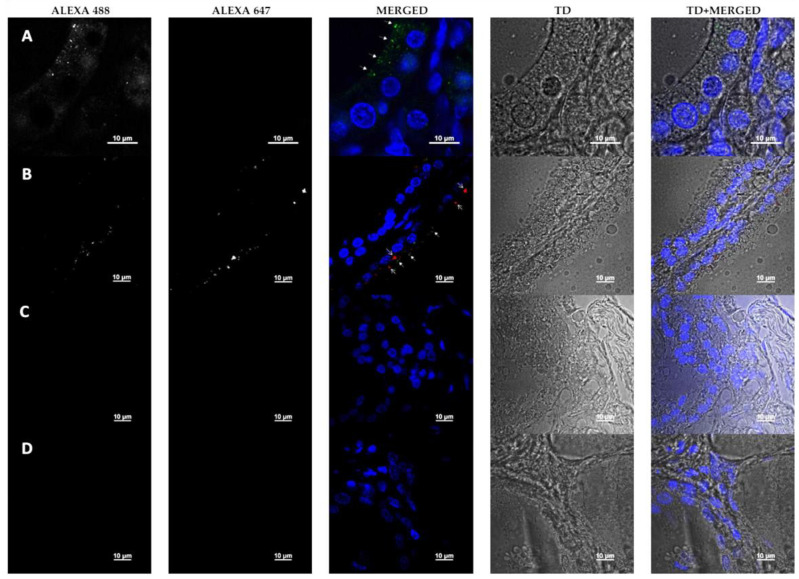
Fluorescent microscopy of thyroid gland tissue. Dyes: DAPI—nucleus; Alexa488—HHV-6 gp82/105 (closed tip arrows (bold)); Alexa647—RANTES (opened tip arrows (sharp)); TD—transmitted light. Channels Alexa488 and Alexa647 are shown in greyscale for better contrast. Row A—part of a thyroid lobule of a patient with Grave’s disease; Row B—limitrophe zone of two separate thyroid follicles of a patient with Grave’s disease; Row C—control patient’s (with adenoma) thyroid lobule; Row D—staining control (secondary antibodies and DAPI only).

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
