# Peer review of "HHV-6 Infection and Chemokine RANTES Signaling Pathway Disturbance in Patients with Autoimmune Thyroiditis"

_viruses, 2020, doi:10.3390/v12060689_

Round 1
Reviewer 1 Report
Sultanova et al.
HHV-6 infection and chemokine RANTES signalling pathway disturbance in patients with autoimmune thyroiditis
Overview:
In this manuscript, Sultanova et al. investigate the association of the virus human herpesvirus-6 (HHV6) with the development of autoimmune thyroiditis (AIT), with a focus on the immunomodulating viral proteins U12 and U51 in AIT and chemokine signaling. 100 patients with AIT were examined for HHV6 DNA, and mRNA for viral genes U79/80, U51 and U12; the latter measure was designed to distinguish latently infected cells from those with active viral replication. The expression of immunomodulatory cytokines (RANTES, IFNg, IL-6 and TNF) was also examined. Control group measures were derived from 30 autopsied subjects without thyroid pathology and 35 healthy blood donors.
Major concerns:
The manuscript, in his current submission, requires improvements in the following three areas: Fig 4 data is incomplete, the writing requires more background and interpretation throughout, and some conclusions are too strong for the data presented.
- Fig 4 requires additional controls/data:
Immunostaining used Abs against viral glycoprotein gp82 and RANTES and demonstrated viral infection in thyrocytes and that HHV6 antigen positive cells often showed RANTES. However, in
Fig 4, I cannot see what is described in the text. The images require a zoomed in version of the panel that is not merged but shows only one channel of interest at a time. The colour should be altered to black and white and only show the channel(s) that are important for better visualization of fine detail. The figure also needs control images from control patients. Several aspects are confusing and should be addressed in the results section. Why is RANTES staining in association with HHV6? Isn’t this contradictory to data that shows lower levels of RANTES in patients with reactivated HHV6? Fig 4 is not interpreted for me. More importantly, data as presented is not enough to support the author’s conclusions.
- Writing/Interpretation:
Background information and rationale:
-Previous studies from this group and others showed clear association between active HHV6 transcription and AIT. It is therefore unclear: what precisely is the goal of this study and what differentiates the present study from what is already know? Please more clearly delineate the knowledge gap that this study addresses.
-Unclear wording “some studies bring evidence of HHV6 in AIT disease. At least one of EBV, CMV, and HHV6 was detected in fresh thyroid gland tissue specimens.” What precisely does this mean?
-More info about viral GPCRs is needed to appreciate their potential role in the development of AIT. What cellular molecules are they homologs of? Are the viral homologs constitutively active and if so, what is the outcome of this constitutive signaling pathway? What might the autoimmune reaction recognize? The discussion only focuses on potential mechanism of RANTES downregulation.
-At no time is the normal role of RANTES in immune function ever described. This information needs to be provided upfront to support the discussion and aid the readers’ interpretation.
In the results section, several areas require addition writing/interpretation:
-HHV6 genome is found in 98% of AIT patient thyroid tissue and in 77% of autopsy control samples. HHV6 genome in 16% of AIT PBMCs. Please interpret these findings for the reader. These tissues that harbour viral DNA could be latent or lytic. Next, HHV6 mRNA for U79/80, U12 and U51 was found in 57% of AIT patients and in 6% of control, while the vGCPR mRNA was only in 39% of AIT patients and not in controls. The presentation of these data as a list makes it confusing and hard to compare. Make a table to accompany Fig 1 for better comparisons. And please interpret more. I am unclear what the significance is of multiple GPCR expression versus just one. And why is the pie chart in Fig 1 broken up? Is there a reason it is graphed this way? I would prefer to see a standard pie chart. Or Venn diagrams to represent overlapping gene expression data. Overall, I am really unsure what you want me to take away from Fig 1 because you did not tell me the important finding. Fig 1 also needs a figure legend describing how the data presented was generated.
-Viral load was determined by nPCR and found to be higher in AIT patient thyroid than control but not significantly so. Why are graphs in Fig 2 colour coded differently from one another?
Labels on the y-axis are also confusing. What do they mean? Does the left panel mean latency AND lytic (DNA detected) and the right panel mean latency only (no mRNA)? Be more specific. This is the type of information that should be in the figure legend, which should be provided.
- The link between RANTES and AIT should be reviewed prior to its examination in the results section. It feels abrupt. AIT patients have less RANTES than control blood donors. In addition, RANTES levels were stratified for patients with active HHV6 (viral mRNA detected) – this group had much lower RANTES than patients with latent infection (not significant). What I liked about this section is that you interpreted the data enough to help me understand that active viral replication was required for the RANTES effect. But what I did not like is that there was no reminder about why I should care what RANTES levels are in AIT patients.
III. Some conclusions are too strong for the data shown:
-The assertion that HHV6 vGPCRs may influence autoimmunity via inhibition of the chemokine RANTES is not directly supported by any data. Soften the conclusion in this regard. The observation that 1/3 of AIT patients express the mRNA for these viral proteins suggests viral reactivation is increased in this subset of patients but is not evidence for causality. In other words, no direct evidence is shown to even suggest that U12 or U51 gene products are responsible for RANTES decreases.
Line 59-60: what is the evidence to suggest that U12 and U51 can interfere? No reference provided. No support provided for how RANTES drop contributes to AIT.
Author Response
- Revisions concerning Fig. 4
- The original version of Fig. 4 was substituted with multiple zoomed in images, showing the different channels – both single channel and merged (page 9);
- 4 caption was rewritten (lines 300-302)
- Zoomed in images, showing the different channels, of control patient sample and staining control were added to Fig. 4 (page 9);
- Results of Fig.4 were rewritten and additional interpretation was added (lines 276-297).
- Revisions made concerning reviewers’ comments about the background information:
- The sentence: “At least one of EBV, CMV, and HHV-6 was detected in fresh thyroid gland tissue specimens from eight autopsies without evidence of active herpesvirus infection” was rephrased to “Study mapping the reservoirs of human herpesviruses, detected EBV, CMV and HHV-6 in thyroid gland tissues of eight autopsies, although without detecting active infection, showing that the thyroid is potentially susceptible to herpesvirus infection.” to make the meaning clearer (lines 50-53) ;
- Additional information concerning viral GPCRs was added – their homology, sequence identity to certain human CCRs (lines 72-76; 78-80);
- Additional information about RANTES normal functions, its role in antiviral immunity, the relationship between RANTES and other autoimmune diseases and the unknown link regarding AIT and RANTES levels was added (lines 85-90);
- Some statements arguing the relevance of this study were added – the lack of knowledge of autoimmunity triggering mechanisms and the poorly studied HHV-6 viral chemokine receptor homologs (lines 91-93).
- Revisions made in the Results section:
- Additional interpretation of results concerning HHV-6 genome detection and gene expression were added as well as Fig. 1 was modified (lines 197-199; 226-228);
- Fig 2. was redone as a dot plot, the confusing coloring was removed and a legend, defining active and latent infection, was added (lines 246-249).
- Revisions concerning conclusions
- Conlusions in both Abstract and Discussion were softened and rephrased (lines 33-35; 382-384; 388).
- The English was revised.
Reviewer 2 Report
Autoimmune disease affects the thyroid more than any other organ, and the incidence of autoimmune thyroiditis (AIT) dramatically increases over time worldwide. The etiological factors of AIT are still unknown. The possible involvement of viruses in the development of different autoimmune diseases has been suggested. In this study, the authors analyzed the involvement of human herpesvirus-6 (HHV-6) in this disease’s development.
100 patients with autoimmune thyroiditis following thyroidectomy were enrolled in this study. The authors detected 98% presence of an HHV-6 genomic sequence in AIT patients' thyroid gland tissues. HHV-6 U79/80, U12 and/or U51 mRNA, which are the markers of active HHV-6 infection were predominant in AIT patients' thyroid tissue samples in comparison with the control group (56% vs 6%). Immunofluorescence analysis detected the HHV-6 antigens in thyrocytes of the patients. RANTES was also detected in thyroid tissues, whereas low level of RNATES was detected in AIT patients' peripheral plasma.
The article is well written, and results suggested the relation between AIT and HHV-6, even though causal relationship of them is not evident. This article will be of broad interest in the readership. However, the reviewer suggests following comments to improve this article.
Specific Comments:
(i) Please indicate 17% U79/80 only, 6% U12 only, and 3% U51 only in the label in fig. 1.
(ii) In 3.2. HHV-6 load in thyroid tissue sample, the authors should indicate 106 instead of 106.
(iii) Please check the label of Fig. 2. Is it really latent infection vs. active infection? In the text, the authors described patient’s vs control. If it is right, the authors should define latent and active in the legend.
(iv) The authors should show dot plot instead of bar chart in the Fig. 2 and 3.
(v) Throughout this paper, this reviewer cannot believe that the authors shows the data of the “role” of HHV-6 in the pathogenesis of AIT. Thus, this reviewer highly recommends that the authors should modified the text such as “These results indicate that a low level
of RANTES in AIT patients can lead to higher activity of HHV-6 at the site of persistence
and this in turn could enhance AIT development.” in the abstract and “these results are indicative of the important role of HHV-6 in AIT development.” in the Discussion.
Author Response
- Revisions made concerning reviewers’ comments about the results section:
- 1 was revised as requested by the reviewer – 17% U79/80 only, 6% U12 only, and 3% U51 only was added to figure legend (page 4);
- HHV-6 viral load was revised as requested by the reviewer – from 106 to 106 (lines 232-242) and results were corrected;
- Additional information defining active and latent infection were added to Fig. 2 legend (lines 246-249);
- Figures 2 and 3 were redone as dot plots;
- Revisions concerning conclusions
Conclusions in both Abstract and Discussion were softened and rephrased (lines 33-35; 382-384; 388).
Round 2
Reviewer 1 Report
The manuscript is significantly improved.
-The addition of additional background information and interpretation of results dramatically increased the clarify of the manuscript.
-Fig 4 is much improved; however a few minor changes should be implemented.Dapi alone panel is not required. Red and green alone panels should be changed to black and white so the staining can be seen better. Arrows or arrowhead symbols should be used to indicate regions of importance.
-One additional point is that figure legends were not improved significantly. Fig 1 and 3 still have no explanation and Fig 2 is exceedingly brief. The figure legend should describe the experiment or procedure in brief so that the reader does not need to go to the methods unless they want additional detail. The authors still need to write adequate figure legends.
-Line 291 - typo "an" important role
Author Response
- Figure 4 was revised- DAPI channel alone was deleted and Alexa488, as well as, Alexa647 channels were shown in grey scale for better contrast. Arrows were used to indicate regions of importance;
- Figures' 1,2 and 3 legends were expanded and explanations were added;
- Line 291 - typo "an" important role was revised.